# The Muscle-Conditioned Medium Containing Protocatechuic Acid Improves Insulin Resistance by Modulating Muscle Communication with Liver and Adipose Tissue

**DOI:** 10.3390/ijms24119490

**Published:** 2023-05-30

**Authors:** Hira Shakoor, Haneen Hussein, Nour Al-Hassan, Mouza Alketbi, Jaleel Kizhakkayil, Carine Platat

**Affiliations:** Department of Nutrition and Health, College of Medicine and Health Sciences, United Arab Emirates University, Al Ain 15551, United Arab Emirates; 201890012@uaeu.ac.ae (H.S.);

**Keywords:** diabetes mellitus, protocatechuic acid, insulin resistance, glucose uptake

## Abstract

Diabetes mellitus is a public health concern, affecting 10.5% of the population. Protocatechuic acid (PCA), a polyphenol, exerts beneficial effects on insulin resistance and diabetes. This study investigated the role of PCA in improving insulin resistance and the crosstalk between muscle with liver and adipose tissue. C2C12 myotubes received four treatments: Control, PCA, insulin resistance (IR), and IR-PCA. Conditioned media from C2C12 was used to incubate HepG2 and 3T3-L1 adipocytes. The impact of PCA was analyzed on glucose uptake and signaling pathways. PCA (80 µM) significantly enhanced glucose uptake in C2C12, HepG2, and 3T3-L1 adipocytes (*p* < 0.05). In C2C12, PCA significantly elevated GLUT-4, IRS-1, IRS-2, PPAR-γ, P-AMPK, and P-Akt vs. Control (*p* ≤ 0.05), and modulated pathways in IR-PCA. In HepG2, PPAR-γ and P-Akt increased significantly in Control (CM) vs. No CM, and PCA dose upregulated PPAR-γ, P-AMPK, and P-AKT (*p* < 0.05). In the 3T3-L1 adipocytes, PI3K and GLUT-4 expression was elevated in PCA (CM) vs. No CM. A significant elevation of IRS-1, GLUT-4, and P-AMPK was observed in IR-PCA vs. IR (*p* ≤ 0.001). Herein, PCA strengthens insulin signaling by activating key proteins of that pathway and regulating glucose uptake. Further, conditioned media modulated crosstalk between muscle with liver and adipose tissue, thus regulating glucose metabolism.

## 1. Introduction

Diabetes mellitus (DM) is a known public health concern. In 2019, the World Health Organization (WHO) ranked diabetes as the ninth leading cause of death [1]. According to the International Diabetes Federation (IDF), the prevalence of diabetes in the United Arab Emirates (UAE) was 11.8% in 2021 compared to 10.5% worldwide [2,3]. Type 2 diabetes mellitus (T2DM) accounts for approximately 90% of all diabetes cases [4] associated with insulin resistance.

Skeletal muscle is among the greatest metabolically active organs in the human body, comprising 40% of total body weight. Due to the mass of skeletal muscle, it is accountable for approximately 80% of postprandial glucose disposal, and therefore plays a pivotal role in the development of insulin resistance and T2DM [5]. When muscle, fat, and liver stop responding properly to insulin, it leads to an impairment of insulin action; this state is known as insulin resistance [6]. Insulin resistance reduces glucose uptake in insulin-sensitive tissues, including skeletal muscle, liver, and adipose tissue [7]. Glucose transport into skeletal muscle is the rate-limiting step for whole-body glucose uptake. Predominantly, skeletal muscle is an important tissue in maintaining glucose homeostasis and is a major site of insulin resistance in T2DM.

In addition to its metabolic activity, skeletal muscle is a secretory organ that releases various cytokines and other peptides known as myokines. Myokines act in an autocrine, paracrine, or endocrine manner and play a key role in regulating lipid mobilization from adipose tissue, liver endogenous glucose production, insulin secretion, glucose disposal, fatty acid oxidation, and lipolysis [8]. Therefore, myokines have a prominent role in maintaining glucose homeostasis. The skeletal muscle secretome comprises several hundred secreted products: interleukin (IL)-6 (IL-6), IL-15, insulin-like growth factor 1 (IGF-1), fibroblast growth factor 21 (FGF-21), myostatin, irisin, and tumor necrosis factor-alpha (TNF-α). They have a considerable impact on glucose homeostasis [9]. It was observed that insulin resistance induced by palmitate in C2C12 cells results in altered muscle secretions (FGF-21, irisin, and myonectin) and glucose transporter 4 (GLUT-4) [10]. Myokines secretion is affected by insulin resistance. This has been revealed by comparing the myokines profile between insulin-resistant and non-insulin-resistant individuals [11]. Insulin resistance may cause the upregulation of IL-6, TNF-α, and myostatin and the downregulation of IL-15, FGF-21, and irisin levels [11].

Polyphenols are secondary metabolites classified as flavonoids, phenolic acids, lignans, and stilbenes [12]. Polyphenols are known for their diverse health benefits, including anti-inflammatory and antioxidative properties, and they improve insulin sensitivity by regulating the insulin signaling pathway and muscle secretions [13]. Flavonoids have especially been shown to improve glucose tolerance in patients with T2DM [14] and insulin sensitivity by activating AMP-activated protein kinase (AMPK) in mice [15]. Protocatechuic acid (PCA), also known as 3,4-dihydroxybenzoic acid, is a hydroxycinnamic acid derivate. There is growing evidence that PCA regulates insulin signaling pathways, e.g., phosphoinositide-3-kinase/Akt (PI3K/Akt) [16]. Activation of PI3K/Akt results in glucose transporter type 4 (GLUT-4) translocation and elevates glucose uptake. Recently, Abdelmageed et al. illustrated that PCA improved hepatic insulin resistance by modulating the insulin receptor substrate 1/phosphoinositide-3-kinase/Akt2 (IRS1/PI3K/Akt2) pathway [17]. PCA enhanced insulin signaling in the rats’ soleus muscle by increasing Akt phosphorylation and mitigating gluconeogenesis [18]. Interestingly, PCA also exerts insulin-like activity by activating peroxisome proliferator-activated receptor gamma (PPAR-γ), a nuclear hormone receptor that controls glucose and lipid metabolism [19]. Moreover, in vivo and in vitro studies have demonstrated that cyanidin-3-O-β-glucoside and PCA activated AMP-activated protein kinase (AMPK) and elevated glucose transporter-1 (GLUT-1) and GLUT-4 expression in the liver, thus improving glucose tolerance [20,21]. It is well established that activation of AMPK in the liver, skeletal muscle, and adipose tissue improves glucose uptake and insulin sensitivity. Therefore, PCA is an important polyphenol, well known for its insulin-like effect, and modulation of insulin signaling pathways could be a potential therapeutic compound to attenuate insulin resistance.

Indeed, polyphenols exert an ability to modulate myokines secretion, thus improving glucose metabolism. For instance, cyanidin-3-glucoside has been shown to elevate glucose uptake in HepG2 cells and C2C12 myotubes and regulate hepatic FGF-21 level [22]. Further, caffeic acid phenethyl ester has been shown to inhibit pro-inflammatory cytokines, i.e., TNF-α and IL-6 [23], resulting in improved insulin sensitivity. To date, research about PCA has been limited in terms of muscle secretions. However, one animal study has shown that the administration of PCA could regulate IGF-1 [24]. Additionally, caffeic acid from coffee suppressed transforming growth factor-beta (TGF-β) and myostatin expression and elevated IGF-1 levels [25].

There is a gap in the literature regarding the impact of polyphenols, especially PCA, on muscle secretions and how they impact muscle communication with the liver and adipose tissue in the presence or absence of insulin resistance. Herein, this study aims to investigate the impact of PCA on insulin resistance by modulating the communication of the muscle with the liver and adipose tissue.

## 2. Results

### 2.1. Study Design

C2C12 cells were differentiated into myotubes. After differentiation, C2C12 myotubes received four treatments: (1) Control; (2) 80 µM of protocatechuic acid (PCA); (3) palmitate (0.2 mM) to induce insulin resistance (IR); and (4) with IR and PCA (IR-PCA). Conditioned media from C2C12 from all four treatments were collected and used to treat HepG2 and 3T3-L1 adipocytes (Figure 1). HepG2 and 3T3-L1 received five treatments: (1) No Conditioned Media (No CM); (2) Control CM; (3) PCA (CM); (4) IR (CM); and (5) IR-PCA (CM). This was aimed at investigating: (1) the effect of PCA on glucose uptake and insulin signaling pathways in C2C12 cells after induction of insulin resistance; and (2) the impact of PCA on glucose uptake and insulin signaling pathways in HepG2 and 3T3L1 in the presence or absence of muscle-conditioned medium after induction of insulin resistance.

### 2.2. Viability of C2C12 Cell for Palmitate

The potential cytotoxic effect of palmitate was investigated, determining its impact on the viability of C2C12 myotubes. Approximately 34% ± 0.30 of cell death was observed at 0.05 and 0.1 mM of palmitate after incubating C2C12 cells for 24 h. It was found that 30% ± 0.23 of cell death occurred at 0.2 mM of palmitate. However, more than 52% ± 0.11 cell death was observed at other doses (0.4, 0.6, 0.8, 1, and 2 mM). Therefore, the 0.2 mM palmitate dose was selected to make cells insulin-resistant (Figure 2).

### 2.3. Viability of C2C12 Cell for PCA

The potential cytotoxic effect of PCA was investigated to assess the viability of C2C12 myotubes. In C2C12, approximately 3% ± 0.09 of cell death was observed at an 80 µM of protocatechuic acid (PCA) dose within 24 h of incubation. Therefore, 80 µM of PCA was selected to treat the cells (Figure 3).

### 2.4. Glucose Uptake by C2C12

C2C12 myotubes were exposed to PCA (80 uM) and palmitate (0.2 mM) to induce insulin resistance (IR) and PCA-IR for 24 h, followed by glucose uptake measurements (Figure 4). A glucose uptake assay was performed in the presence and absence of insulin. However, glucose uptake in PCA was 95.63 ± 3.9 pmol, in IR was 69.73 ± 3.51 pmol, and in IR-PCA was 76.62 ± 0.78 pmol without insulin stimulation. However, in PCA, a significant increase in glucose uptake (133.25 ± 11.30 pmol) was observed compared to IR (79.24 ± 7.99 pmol) (*p* < 0.05) with insulin stimulation. Additionally, glucose uptake was greater in IR-PCA (87.85 ± 6.72 pmol) than in IR, but the results are not statistically significant.

### 2.5. Glucose Uptake by HepG2

HepG2 cells were exposed to conditioned media (Control, PCA, IR, and IR-PCA) for 24 h, followed by glucose uptake measurements (Figure 5). A glucose uptake assay was performed in the presence and absence of insulin. Glucose uptake in Control (CM) was 24.13 ± 3.12 pmol with insulin stimulation and 15.58 ± 0.97 pmol without insulin stimulation. Interestingly, by treating cells even with Control (CM), the glucose uptake was significantly increased compared to No CM: 95.09 ± 0.39 pmol with insulin and 98.33 ± 2.83 pmol without insulin (*p* ≤ 0.001). In PCA (CM), there was a drastic increase in glucose uptake rate to 150.82 ± 3.9 pmol with insulin stimulation, which was significantly different from other treatments (*p* ≤ 0.05). In IR-PCA (CM), glucose uptake was 133.32 ± 17.64 pmol with insulin stimulation. However, glucose uptake was significantly greater in IR compared to No CM with/without insulin stimulation (*p* = 0.002), which could be because of beneficial muscle secretions present in conditioned media.

### 2.6. Glucose Uptake by 3T3-L1 Adipocytes

Next, 3T3-L1 adipocytes were exposed to conditioned media (Control, PCA, IR, and IR-PCA) for 24 h, followed by glucose uptake measurements (Figure 6). A glucose uptake assay was performed in the presence and absence of insulin. Glucose uptake increased in Control (CM) (205.32 ± 8.38 pmol) in the presence of insulin compared to No CM (146.90 ± 0.01). In PCA (CM), glucose uptake increased significantly to 264.29 ± 20.66 pmol without insulin and 326.02 ± 20.66 pmol with insulin stimulation (*p* ≤ 0.05). Moreover, a significant elevation in glucose uptake was observed in IR-PCA (CM), to 213.58 ± 2.73 pmol without insulin and 289.37 ± 24.75 pmol with insulin, compared to IR (CM), with 105.90 ± 8.09 pmol without insulin and 170.77 ± 91.73 pmol with insulin (*p* ≤ 0.05).

### 2.7. Effect of Protocatechuic Acid on Insulin Signaling in C2C12 Myotubes

The protein expression level of GLUT-4, PI3K, IRS-1, IRS-2, PPAR-γ, Akt, *P*-Akt, AMPK, and P-AMPK was assessed by Western blotting in C2C12 myotubes (Figure 7A). Protocatechuic acid (PCA) significantly stimulated the expression of GLUT-4 by 78% ± 0.15 (*p* < 0.05) (Figure 7C). GLUT-4 expression improved significantly in IR-PCA by 238.65% ± 0.14 compared to IR (*p* < 0.05) (Figure 7C). PCA activated PI3K expression by 57% ± 3.94 vs. Control and IR-PCA by 64% ± 3.28 vs. IR but did not reach statistical significance (Figure 7B). Additionally, IRS-1 expression increased by 7% ± 0.01 in PCA compared to IR, while IR-PCA was significantly increased by 63% ± 0.02 compared to IR (*p* < 0.001) (Figure 7D). IRS-2 expression was significantly increased by 52% ± 0.25 compared to IR (*p* < 0.001) (Figure 7E). PPAR-γ expression was significantly increased in PCA by 1.7-fold vs. Control and in IR-PCA by 1.7-fold vs. IR (*p* < 0.001) (Figure 7F).

Protocatechuic acid (PCA) significantly elevated the P-AMPK expression compared to Control (*p* = 0.02) and PCA vs. IR (*p* = 0.01) (Figure 8A,B). It was observed that in PCA, P-Akt was significantly raised by 2.6-fold and in IR-PCA by 4-fold compared to Control (*p* ≤ 0.001) (Figure 9A,B).

### 2.8. Effect of Protocatechuic Acid on Insulin Signaling in HepG2 Cells

The level of protein expression of GLUT-4, PI3K, IRS-1, IRS-2, PPAR-γ, Akt, P-Akt, AMPK, and P-AMPK was assessed by Western blotting in HepG2 cells (Figure 10A). HepG2 cells were treated with conditioned media, i.e., Control (CM), PCA (CM), IR (CM), and IR-PCA (CM). In Control (CM), there was upregulation of PI3K by 19% ± 1.65 and IRS-1 by 137% ± 6.20 compared to No CM (Figure 10B). PCA remarkably improved the expression of PI3K, GLUT4, IRS-1, and IRS-2 (Figure 10A–E). However, a significant elevation in GLUT-4 was found in PCA (CM) compared to IR (CM) (*p* < 0.05). A notable improvement was observed in the band intensity of PI3K by 0.5-fold, GLUT-4 by 1.4-fold, IRS-1 by 0.6-fold, and IRS-2 by 0.3-fold in IR-PCA vs. IR. The expression of PPAR-γ was significantly enhanced in Control (CM) by 1.4-fold compared to No CM (*p* < 0.05). Further, PPAR-γ protein expression was significantly increased in PCA (CM) by 0.8-fold compared to No CM (*p* = 0.005). In contrast, the PPAR-γ protein expression was significantly decreased in IR-PCA (CM) (*p* < 0.05), as shown in Figure 10F.

It was observed that P-AMPK expression increased by 20% ± 0.5 in Control (CM) and further increased in IR-PCA (CM) by 2.2-fold compared to IR (*p* ≤ 0.01). Interestingly, the increase in P-AMPK expression in IR-PCA (CM) was greater than in PCA (CM) (Figure 11A,B) A similar trend was observed in P-Akt expression; there was a significant increase in the expression of P-Akt in Control (CM) and PCA (CM), while the increase in IR-PCA (CM) was greater than in other treatments (Figure 12A,B). The increase in P-Akt expression in Control (CM) and PCA (CM) was 78% ± 0.09 and 46% ± 0.09, respectively (*p* ≤ 0.001), compared to No CM. In addition, in IR-PCA (CM), P-Akt expression increased significantly by 4-fold compared to IR (*p* ≤ 0.001).

### 2.9. Effect of Protocatechuic Acid on Insulin Signaling in 3T3-L1 Adipocytes

The level of protein expression of GLUT-4, PI3K, IRS-1, IRS-2, PPAR-γ, Akt, P-Akt, AMPK, and P-AMPK was assessed by Western blotting in 3T3-L1 adipocytes (Figure 13A). Then, 3T3-L1 adipocytes were treated with conditioned media, i.e., Control (CM), PCA (CM), IR (CM), and IR-PCA (CM). In Control (CM), expression of PI3K increased by 44% ± 0.01, GLUT-4 by 252% ± 0.09, and IRS-2 protein by 74% ± 0.94 vs. No CM. In PCA (CM), PI3K expression increased significantly (*p* < 0.05) (Figure 13B). It was found that GLUT-4 expression was significantly greater in PCA (CM) compared to other groups (*p* < 0.001) (Figure 13C).

There is a significant decreasing trend in IRS-1 expression in Control (CM) in 3T3-L1 adipocytes (*p* < 0.01) (Figure 13D). However, the IRS-1 expression was significantly greater in PCA and IR-PCA, by 11-fold and 7.6-fold, respectively, compared to IR (*p* < 0.001). It was found that the expressions of PI3K, GLUT-4, and IRS-2 increased in IR-PCA (CM) compared to IR (Figure 13E). However, no significant difference was found in PPAR-γ protein expression among all groups treated with conditioned media in 3T3-L1 adipocytes (Figure 13F).

P-AMPK expression decreased significantly in IR (CM) compared to all other groups by treating 3T3-L1 adipocytes with conditioned media (*p* ≤ 0.01) (Figure 14A,B). In contrast, treatment of 3T3-L1 adipocytes with Control (CM), PCA (CM), and IR-PCA (CM) did not cause significant change. Furthermore, an increase in P-Akt was seen in Control (CM), PCA (CM), and IR-PCA (CM) but the results are not statistically significant (Figure 15A,B).

## 3. Discussion

It is evident that skeletal muscle acts as an endocrine organ, secreting myokines that communicate with various other metabolic organs, including the liver and adipose tissues, thus regulating energy metabolism. Myokines (irisin, IL-6, IL-10, myonectin, secreted protein acidic rich in cysteine (SPARC), β-aminoisobutyric acid (BAIBA), and FGF-21) crosstalk with adipose tissue to increase glucose uptake and improve insulin sensitivity [26]. Interestingly, patients with T2DM had shown a low level of beneficial myokines, i.e., irisin, IL-13, and Follistatin-like 1 (FSTL-1) [27]. Similarly, another study demonstrated that isolating human primary skeletal muscle cells from a patient with T2DM had altered myokine profiles compared to healthy human skeletal muscle cells [28]. This indicates that insulin resistance and diabetes negatively affect myokines secretion.

It is well established that myokines have a potential role in improving glucose uptake by modulating insulin signaling pathways. For instance, FSTL-1 (myokine) stimulates muscle glucose uptake by activating AMPK [29,30]. Further, ex vivo studies have shown that the culture of mouse and human islets with muscle-derived, irisin-enriched conditioned media from palmitate-treated L6 myotubes leads to insulin biosynthesis and protects β-cell apoptosis induced by palmitate [31]. Moreover, palmitate-exposed C2C12 myocytes, when treated with BAIBA (myokine), improved defected insulin receptor substrate 1 (IRS)-1/Akt, phosphorylation of AMPK, and PPAR-γ expression [32]. Evidence showed that treating C2C12 myotubes with IL-15 results in the elevation of GLUT-4 gene expression and translocation of GLUT-4 vesicles [33,34]. Activating PI3K/Akt in C2C12 myotubes increased FGF-21 expression [35]. Therefore, the present study was conducted to analyze the impact of conditioned media from C2C12 cells containing myokines and their impact on glucose metabolism.

PI3K has an important role in insulin signaling. It stimulates phosphoinositide phosphorylation to produce phosphatidylinositol-3,4,5-phosphates, hence translocating glucose transporter and leading to glycogen synthesis [36,37]. The present study found that PCA upregulated PI3K expression in C2C12 myotubes. Moreover, PCA attenuated the negative effect of palmitate and upregulated PI3K by 0.6-fold in IR-PCA vs. IR. Our findings follow the previous results of Chen et al., who demonstrated that 100 μg/mL of chlorogenic acid and caffeic acid increased PI3K expression by 66.7% in HepG2 cells [38]. Another study illustrated that young apple polyphenols and their two main compounds, i.e., phlorizin and chlorogenic acid, alleviated insulin resistance in HepG2 cells and activated the PI3K/AKT pathway [39]. A similar trend was observed when HepG2 and 3T3-L1 adipocytes were incubated with conditioned media for 24 h. Interestingly, PI3K was activated in HepG2 and 3T3-L1 adipocytes by 0.1- and 0.4-fold, respectively, by incubating cells only with Control (CM); this improvement could be due to muscle cell secretions.

GLUT-4 is a glucose transporter belonging to the GLUT family, widely distributed in endosomes, trans-Golgi networks, and tubular–vesicular structures. A defect in the translocation of GLUT-4 could lead to insulin resistance, type 2 diabetes, and metabolic syndrome [40]. We used Western blotting to investigate the potential effect of PCA on the expression of GLUT-4. In C2C12 myotubes, GLUT-4 expression increased by 0.7-fold with PCA (80 μM) treatment compared to Control and 2.3-fold in IR-PCA compared to IR. Our findings are in accordance with Son et al.’s results, which demonstrated that aspalathin (flavonoid) modulated glucose uptake and activated P-AMPK, thus resulting in the upregulation of GLUT-4 translocation to the membrane in L6 myotubes [41]. Interestingly, conditioned media containing PCA significantly improved the GLUT-4 expression in the present findings (*p* < 0.05). When HepG2 and 3T3-L1 adipocytes were treated with PCA (CM), the GLUT-4 expression was elevated by 0.6- and 2.5-fold, respectively, compared to Control. Moreover, GLUT-4 expression in HepG2 and 3T3-L1 adipocytes increased by 1.4- and 3.4-fold, respectively, compared to IR. Previously, it was found that caffeic acid mitigated insulin resistance and enhanced glucose uptake in HepG2 cells [38]. Scazzocchio et al. reported that PCA (100 μM) treatment to murine adipocyte 3T3-L1 cells significantly increased glucose uptake and GLUT-4 expression [21]. However, in HepG2 cells, glucose uptake increased with the control-conditioned medium, but no change was observed in GLUT-4 expression. Increased glucose uptake in liver cells might be due to the GLUT-2 transporter, which is highly expressed in the liver. In contrast, the GLUT-4 transporter is mainly expressed in muscle and adipose tissue [40].

Further, in HepG2, glucose uptake was significantly greater in the presence or absence of insulin in the IR group than in the No CM, and in 3T3-L1 adipocytes, glucose uptake was significantly increased in the presence of insulin in the IR than in No CM. This suggests that muscle secretions in conditioned media may attenuate the negative impact of palmitate.

Initiation of insulin signal transduction occurs when insulin binds to the insulin receptor (IR), which stimulates various intracellular protein substrates such as insulin receptor substrate (IRS)-1 and IRS-2, activating the P13K pathway that results in the stimulation of AKT. In C2C12 myotubes, IRS-1 expression increased by 7% ± 0.01 in PCA vs. Control and by 63% ± 0.02 in IR-PCA vs. IR (*p* ≤ 0.001). In HepG2, IRS-1 expression increased by 2.2-fold in PCA (CM) compared to Control and by 0.6-fold in IR-PCA (CM) compared to IR (CM). Further, in 3T3-L1 adipocytes, a notable increase in IRS-1 expression was observed by 7.6-fold in IR-PCA (CM) compared to IR (CM). It has previously been shown that a polyphenol named silibinin prevented IRS-1/PI3K/Akt inhibition in C2C12 myotubes, hence preventing insulin resistance induced by palmitate [42]. Furthermore, a significant elevation of IRS-2 expression in IR-PCA vs. IR was observed in C2C12 myotubes. It was noted that IRS-2 expression was upregulated in HepG2 and 3T3-L1 adipocytes by 0.3- and 0.1-fold, respectively, in Control (CM) vs. No CM. This leads to the fact that conditioned media from C2C12 myotubes might have some beneficial secretions modulating the IRS-1 expression. In HepG2 and 3T3-L1 adipocytes, IRS-2 expression increased by 0.7-fold in both cell lines with PCA vs. Control, while it was enhanced by 0.3- and 1.3-fold in IR-PCA vs. IR, respectively. Consistent with our results, a study by Cordero-Herrera et al. reported that epicatechin and cocoa phenolic extracts possess an insulin-like activity by enhancing total levels of IRS-1 and IRS-2, activating the PI3K/AKT pathway and increasing AMPK expression in HepG2 cells [43]. Similarly, another study indicated that in preadipocytes, grape-seed procyanidin extract enhanced IR and IRS levels [44].

Among the diabetes-related signaling molecules, PPAR-γ (a transcription factor) helps to regulate lipid and glucose metabolism, enhancing insulin sensitivity [45,46]. PPAR-γ expression was significantly increased in PCA vs. Control and IR-PCA vs. IR (*p* < 0.001) in C2C12. Similar results were observed in another study in murine adipocyte 3T3-L1 cells treated with PCA [21]. Incubation of HepG2 with conditioned media in Control (CM) and PCA (CM) increased PPAR-γ expression but decreased it in IR-PCA (CM). However, in 3T3-L1, there was no significant difference among the different treatments of conditioned media. A study has shown a decrease in PPAR-γ mRNA expression in lipid-laden 3T3-L1 adipocytes when treated with the polyphenol curcumol [47].

AMPK is an energy receptor that helps to regulate energy metabolism in skeletal muscle, adipocytes, liver cells, and endothelial cells [48]. There was an increase in P-AMPK expression in PCA by 2.7-fold vs. Control, and in IR-PCA, expression of P-AMPK increased to 3-fold vs. IR. A study showed that palmitate-induced insulin resistance in C2C12 myotubes, when treated with Platycodon grandiflorum seeds enriched with luteolin, significantly activated insulin-independent AMPK and GLUT-4 translocation by 1.7-fold [49]. Cyanidin-3-O-β-glucoside (C3G) and protocatechuic acid (PCA) activated AMPK in the murine hepatic cell line [20]. Bioactive compounds such as agrimonolide and desmethylagrimonolide significantly reversed the AMPK phosphorylation in insulin-resistant HepG2 cells compared with control cells [50]. These previous findings are consistent with the findings of the present study.

Akt (a serine/threonine protein kinase) is involved in the translocation of GLUT-4 to the plasma membrane by activating signaling cascades [51]. In the current study, PCA treatment upregulated the expression of P-Akt vs. Control in all cell lines (C2C12, HepG2, 3T3-L1). The P-Akt expression increased when HepG2 and 3T3-L1 adipocytes were treated with conditioned media. Nevertheless, the positive effect could be because of the beneficial myokines present in conditioned media. Our results are in accordance with the finding of Dai et al., who showed that quercetin mitigated insulin resistance induced by tumor necrosis factor-alpha (TNF-α) by enhancing basal and insulin-stimulated glucose uptake via Akt and AMPK activation in C2C12 myotubes [52]. In line with this, another study in C2C12 myotubes reported that 25 µM of Chicoric acid significantly increased glucose uptake and P-Akt level independent of insulin [53]. Then, 3T3-L1 adipocytes incubated with conditioned medium from lipopolysaccharide-stimulated RAW264.7 cells, when treated with resveratrol, improved insulin-stimulated Akt phosphorylation [54].

In the present study, C2C12 myotubes exposed to palmitate (0.2 mM) resulted in dysregulation of glucose metabolism. Additionally, alteration in glucose uptake and transport might be linked with increased lipid storage and decreased glycerol release. It was found that palmitate exposure to cells downregulated P-Akt protein expression, whereas it slightly upregulated P-AMPK [55,56]. Besides regulating glucose uptake, P-AMPK is concomitant with increasing control of energy production via β-oxidation [56]. Indeed, AMPK acts as an energy sensor that responds to cellular energy demands regularly by sensing the balance in the AMP: ATP ratio [57,58]. Thus, AMPK activation could be because of decreased ATP levels induced by palmitate, while PCA attenuated the negative impact of palmitate in IR-PCA. A significant increase in P-Akt and P-AMPK was found in IR-PCA (CM) vs. IR in HepG2 cells and 3T3-L1 adipocytes (*p* < 0.05). On the other hand, treatment with PCA alone decreased P-AMPK levels compared to No CM and Control (CM), but the result did not reach a significant level.

It is well established that skeletal muscle communicates with the liver and adipose tissue via its secretory products. The current study showed an improvement in glucose uptake and glucose metabolism in liver and adipose cells when incubated with conditioned medium. There might be some myokines present in a conditioned media that help in the modulation of glucose metabolism via communication with the liver and adipose tissue. For instance, some myokines crosstalk with other organs, including IL-6, which is involved in glucose metabolism via communication between the muscle and the liver and adipose tissue by enhancing AMPK and Akt expressions [59,60]. IL-15 plays a crucial role in muscle–adipose tissue interaction and is involved in lipid metabolism [61]. Additionally, muscle-derived FGF-21 exerts metabolic actions on the liver, thereby improving hepatic insulin sensitivity [62]. Increased myostatin results in decreasing tissue growth factors that cause a reduction of muscle mass, which leads to a reduction in the sensitivity of hepatocytes to insulin by decreasing IRS-1 and IRS-2 expressions [63]. Recent evidence has shown that polyphenols positively modulate myokines secretion, thereby improving glucose metabolism [64,65,66,67]. Similarly, the current study’s findings showed that conditioned media containing muscle secretions play a beneficial role in improving glucose uptake and signaling pathways. In addition, PCA treatment further enhanced the positive potential of muscle secretions. Therefore, modulating muscle crosstalk with the liver and adipose tissue ameliorates the negative impact of insulin resistance, thus improving glucose metabolism. However, further research needs to be conducted regarding the impact of polyphenols on myokine secretions and their impact on inter-organ communication in reversing insulin resistance.

## 4. Materials and Methods

### 4.1. C2C12 Culture

C2C12 cells (American Type Culture Collection, Manassas, VA, USA) were seeded in a (100 mm) tissue culture plate at a density of 2.5 × 10^5^ cells/well, with a 10 mL growth medium consisting of Dulbecco’s Modified Eagle Medium (DMEM; 4500 mg/mL glucose; Sigma Aldrich, St. Louis, MO, USA), supplemented with 10% fetal bovine serum (FBS; BioWest, Nuaillé, France) and 1% penicillin–streptomycin (PS; Invitrogen, Waltham, MA, USA). The cells were maintained in an incubator at 37 °C and 5% CO_2_. Upon reaching 80% confluence, the medium was changed to a differentiation medium that consists of DMEM supplemented with 2% horse serum (Sigma Aldrich, St. Louis, MO, USA). The cells were used for experiments on day 7 post-differentiation.

C2C12 myotubes received four treatments after differentiation: (1) Control (without insulin resistance and without PCA); (2) PCA (without insulin resistance and with PCA); (3) IR (with insulin resistance and without PCA); (4) IR-PCA (with insulin resistance and with PCA).

### 4.2. HepG2 Liver Cells Culture

HepG2 (American Type Culture Collection, Manassas, VA, USA) was cultured in 75 cm^2^ cell culture flasks, at 37 °C, in a humidified atmosphere of 5% CO_2_/95% O_2_, at a seeding density of approximately 10^5^ cells/cm^2^, in DMEM, and supplemented with 10% FBS and 1% penicillin–streptomycin. Cells grew until 80–90% confluence.

### 4.3. T3-L1 Pre-Adipocytes Culture

3T3-L1 pre-adipocytes (American Type Culture Collection, Manassas, VA, USA) were cultured in DMEM containing 10% FBS and 1% penicillin–streptomycin. Differentiation was induced on day 4 following confluence by replacing the medium with DMEM containing 10 μg/mL insulin, 2.5 μM dexamethasone (Sigma Aldrich), and 0.5 mM 3-isobutyl-1-methylxanthine (Sigma Aldrich) (day 0). Two days later, the medium was replaced with DMEM containing 10% FBS, 10 μg/mL insulin, and 1% penicillin–streptomycin and changed every two days thereafter. The cells were used for experiments 4–10 days after differentiation.

### 4.4. Cell Viability Test

C2C12 cells were cultured in the 96-well plate (1 × 10^4^ cells/well) and overnight incubation was performed. After 24 h, they were treated with various concentrations (µM) at 20, 40, 60, 80, 100, 200, and 500 µM of protocatechuic acid (PCA) for 24 h. The cells were also treated with palmitate (to induce insulin resistance) at concentrations of 0.05, 0.1, 0.2, 0.4, 0.6, 0.8, 1, and 2 mM for 24 h. Then, 25 μL of 3-(4,5-dimethylthiazol-2-yl)-2,5-diphenyltetrazolium bromide (MTT) (Abcam) with a concentration of 5 mg/mL was added to each well and incubated for 3 h at 37 °C. The MTT solution was removed. Then, 200 µL of dimethyl sulfoxide (DMSO; Sigma Aldrich) was added to dissolve the formazan crystals. After mixing the DMSO and formazan crystals well, the absorbance was measured at 570 nm via a microplate reader (Multiscan Go, Thermo-Fisher Scientific, Waltham, MA, USA). All experiments were performed in quadruplicate.

### 4.5. C2C12 Myotubes Insulin Resistance Model

C2C12 on days 7–8, when the cells convert to myotubes, were treated with palmitate. C2C12 myotubes were treated with palmitate (Chem Cruz, Dallas, TX, USA) at 0.2 mM for 24 h of chronic treatment to induce insulin resistance. Palmitate was prepared by diluting in absolute ethanol (80 mM stock solution) and subsequently adding DMEM with 10% FBS and 20% BSA to the ethanol solution (final stock solution concentration of 3.2 mM) so that palmitic acid conjugates with albumin and becomes water-soluble. After 24 h, the media was collected and stored at −80 °C.

### 4.6. PCA Treatment

C2C12 myotubes were treated with 80 µM of protocatechuic acid (PCA) (97% pure) (Sigma Aldrich, St. Louis, MO, USA) for 24 h. PCA was dissolved in dimethyl sulfoxide (Sigma Aldrich). After 24 h, media was collected and stored at −80 °C.

### 4.7. Treatment of 3T3-L1 Adipocytes and HepG2 cells with C2C12 Conditioned Media

Conditioned media from C2C12 with/without insulin resistance and with/without PCA were collected. Other cell lines, including HepG2 and differentiated 3T3-L1 adipocytes, were incubated with conditioned mediums for 24 h. Only 50% of conditioned mediums were used to culture 3T3-L1 adipocytes and HepG2 cells.

### 4.8. Glucose Uptake

Glucose uptake was assayed according to the established protocol from a commercially available glucose uptake kit (ab136955; Abcam, Cambridge, UK) in C2C12 myotubes, 3T3-L1 adipocytes, and HepG2. C2C12 cells were seeded in a 96-well plate (1 × 10^4^ cells/well) and matured into myotubes. After maturation, they were treated with PCA, IR, and IR-PCA for 24 h. After 24 h, the media was replaced with a serum-free DMEM medium and incubated overnight. In addition, HepG2 cells were seeded in 96-well plates (1 × 10^4^ cells/well) and treated with 50% conditioned media (PCA, IR, and IR-PCA treatment) and 50% DMEM media for 24 h. Then, 3T3-L1 were seeded into a 24-well plate (1 × 10^5^ cells/well), after differentiation into adipocytes treated with 50% conditioned media and 50% DMEM media for 24 h. Glucose-free media was added with an incubation time of 40 min. Subsequently, cells were stimulated with 1 μM insulin (Sigma Aldrich, St. Louis, MO, USA) for 20 min. An amount of 10 mM 2-deoxyglucose was added and incubated for an additional 20 min. Cells were washed 2× with cold PBS buffer, lysed with extraction buffer, frozen at −80 °C for 10 min, and heated at 85 °C for 40 min. After cooling on ice for 5 min, the lysates were neutralized by adding a neutralization buffer. Centrifugation was performed and the remaining lysate was diluted with assay buffer. Finally, the end product was produced with an amplification step per the kit protocol. Absorbance was measured at 412 nm using a microplate reader (Multiscan Go, Thermo-Fisher Scientific, Waltham, MA, USA). All experiments were performed in duplicate.

### 4.9. Protein Expression

C2C12 myotubes were treated with PCA, IR, and IR-PCA. However, 3T3-L1 adipocyte and HepG2 cells were treated with conditioned media collected from C2C12 myotubes. All three cells, including C2C12 myotubes, 3T3-L1 adipocyte, and HepG2, were washed with phosphate buffer saline (PBS). Cells were harvested using a cell scraper into a fresh tube with 200 μL of RIPA cell lysis buffer (Sigma Aldrich, St. Louis, MO, USA) containing 1% protease phosphatase inhibitor cocktail (Sigma Aldrich, St. Louis, MO, USA), 1mm phenyl methyl sulfonyl fluoride (PMSF), and 10 mm dithiothreitol (DTT) added to the cells. Cell lysates were centrifuged at 16,500× *g* for 15 min at 4 °C. The total amount of protein was determined by Bio-Rad protein assay (Hercules, CA, USA), diluted with 6× loading buffer, and boiled at 90 °C for 5 min. Loaded 35 μg/lane of protein samples, separated by SDS–PAGE, were transferred onto a nitrocellulose membrane by wet transfer using a Bio-Rad Electrotransfer apparatus. The blocking of membranes was achieved by 5% non-fat milk in tris-buffered saline containing 0.1% Tween 20 (TBS-T) for 1 h at room temperature and immunoblotted using polyclonal primary antibodies against GLUT-4 (1F8), AMPK, P-AMPK (T172), Akt, P-Akt (S473), PI3K (p85), IRS-1, IRS-2, and PPAR-γ (81B8) antibodies (Cell Signaling Technology, Danvers, MA, USA). The membranes were incubated overnight with primary antibodies. The appropriate horseradish peroxidase-conjugated secondary antibodies (anti-rabbit IgG and anti-mouse IgG) (Jackson Immune Research, Cambridge House, UK) were introduced to the blot for 1 h after washing the blot with TBS-T 4×. The band densities were detected using an enhanced chemiluminescence detection kit (Bio-Rad, Hercules, CA, USA). The band densities were quantified using an image analyzer Quantity One System (Bio-Rad). All experiments were performed in duplicate.

### 4.10. Statistical Analysis

Statistical analysis was performed using GraphPad Prism 9.0 (GraphPad, San Diego, CA, USA). All values were represented as Mean ± S.D. The statistical significance of experimental observations was determined using analysis of variance (ANOVA), followed by Dunnett (Figure 1 and Figure 2) by comparing the treatment with Control. However, comparisons between groups were analyzed by ANOVA with Tukey’s post hoc analysis. In all analyses, *p* ≤ 0.05 was considered statistically significant (* *p* ≤ 0.05).

## 5. Conclusions

PCA has shown a therapeutic role in improving glucose uptake, modulating signaling pathways, and attenuating insulin resistance. PCA possesses an insulin-like activity, as it enhances the total levels of IRS-1 and IRS-2 and activates the PI3K and P-Akt expressions in muscle cells, as shown in Figure 16. Moreover, there is an increased glucose uptake and upregulation of GLUT-4 and P-AMPK expression in C2C12 myotubes. Nevertheless, PCA reverses the negative impact of palmitate-induced insulin resistance.

The skeletal muscle is an important secretory organ of the body that is involved in regulating glucose metabolism via its secretions. Incubating HepG2 cells and 3T3-L1 adipocytes in Control CM resulted in upregulation of glucose uptake and modulation of the signaling pathway compared to No CM. The present study has shown that the myokines in conditioned media were involved in crosstalk with the liver and adipose tissue to improve glucose metabolism. Further, conditioned media with PCA from C2C12 cells enhanced glucose uptake, positively modulated insulin signaling pathways, and reversed insulin resistance in HepG2 and 3T3-L1 adipocytes. It is suggested that the myokines in conditioned media might contribute to improving glucose metabolism, leading to an important novel therapy for insulin resistance and T2DM.

## Figures and Tables

**Figure 1 ijms-24-09490-f001:**
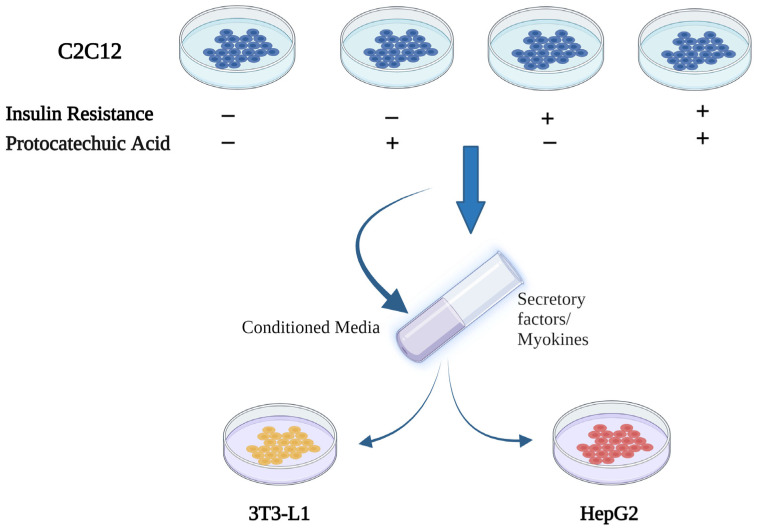
The study design.

**Figure 2 ijms-24-09490-f002:**
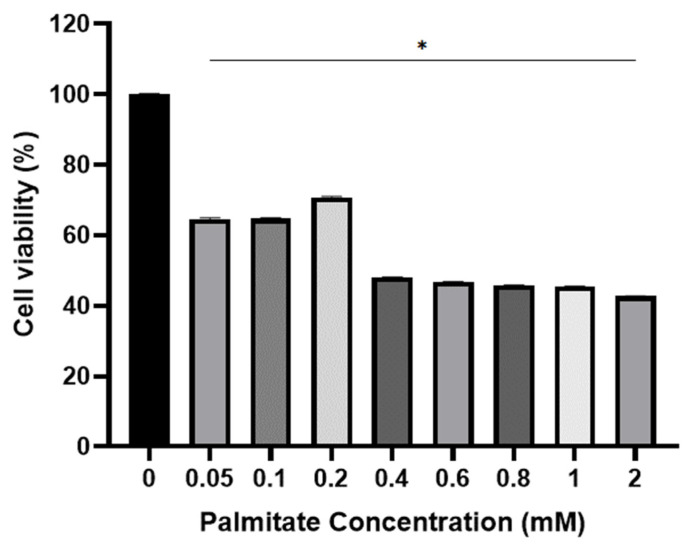
The effect of palmitate-induced insulin resistance on C2C12 cell viability. ANOVA was used to compare Control (0 mM) and other concentrations. Values are the Mean ± S.D. calculated from four replicates. * *p* ≤ 0.05 indicates significantly different from 0 mM of palmitate.

**Figure 3 ijms-24-09490-f003:**
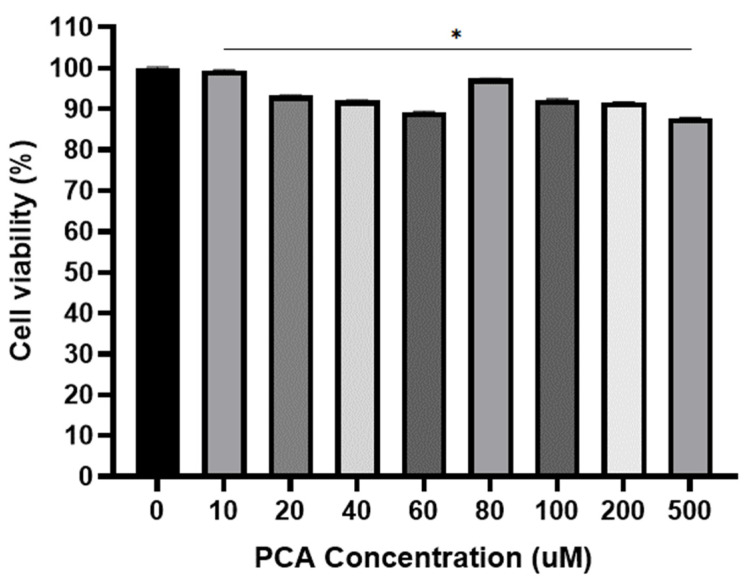
The effect of PCA on C2C12 cell viability. ANOVA was used to compare Control (0 uM) and other concentrations. Values are the Mean ± S.D. calculated from four replicates. * *p* ≤ 0.05 indicates significantly different from 0 mM of PCA.

**Figure 4 ijms-24-09490-f004:**
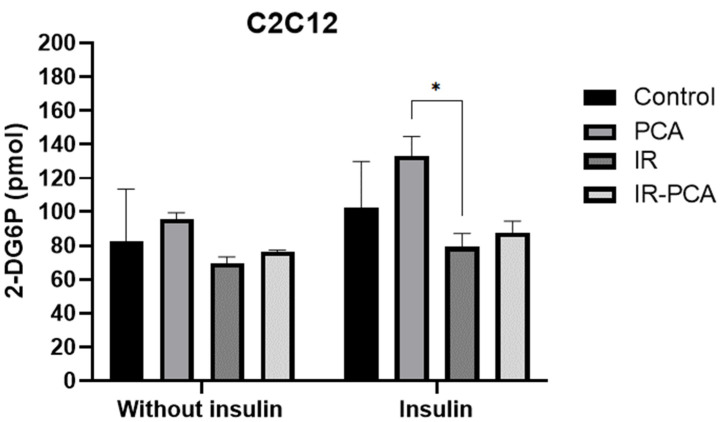
The glucose uptake rate of C2C12 cells in Control, protocatechuic acid (PCA), insulin resistance (IR), and insulin resistance-protocatechuic acid (IR-PCA), without insulin and with insulin-stimulated conditions. Values are the Mean ± S.D. of the duplicate experiment. ANOVA with multiple comparisons was performed and statistical significance was set at *p* ≤ 0.05. * Statistically significant (*p* ≤ 0.05).

**Figure 5 ijms-24-09490-f005:**
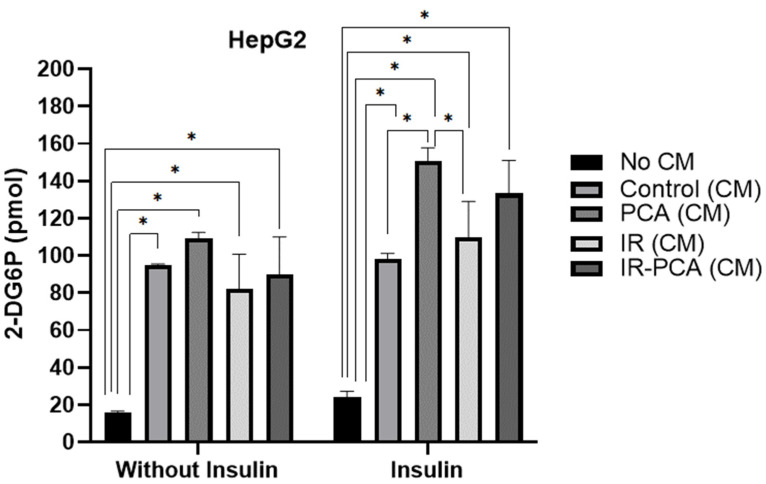
The glucose uptake rate by human liver cells HepG2 of no conditioned media (CM), Control (CM), PCA (CM), IR (CM), and IR-PCA (CM), without insulin and with insulin-stimulated conditions. Values are the Mean ± S.D. of the duplicate experiment. ANOVA with multiple comparisons was performed and statistical significance was set at *p* ≤ 0.05. * Statistically significant (*p* ≤ 0.05). Conditioned Media (CM).

**Figure 6 ijms-24-09490-f006:**
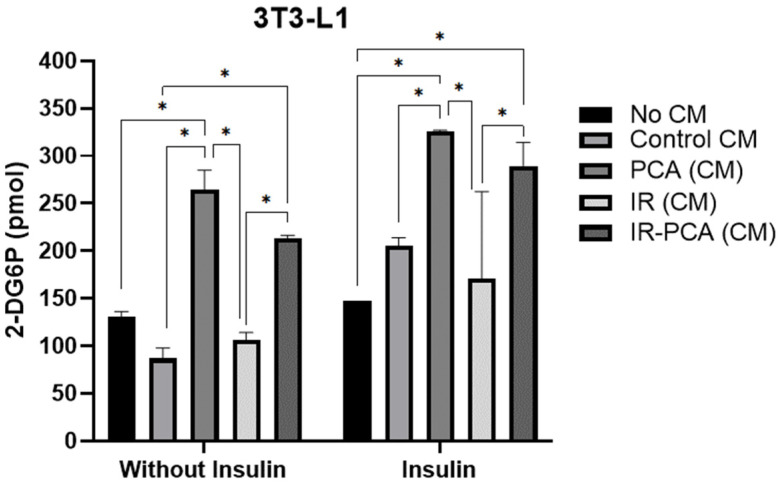
The glucose uptake rate by 3T3-L1 adipocytes of no conditioned media (CM), Control (CM), PCA (CM), IR (CM), and IR-PCA (CM), without insulin and with insulin-stimulated conditions. Values are the Mean ± S.D. of duplicate experiments. ANOVA with multiple comparisons was performed and statistical significance was set at *p* ≤ 0.05. * Statistically significant (*p* ≤ 0.05). Conditioned Media (CM).

**Figure 7 ijms-24-09490-f007:**
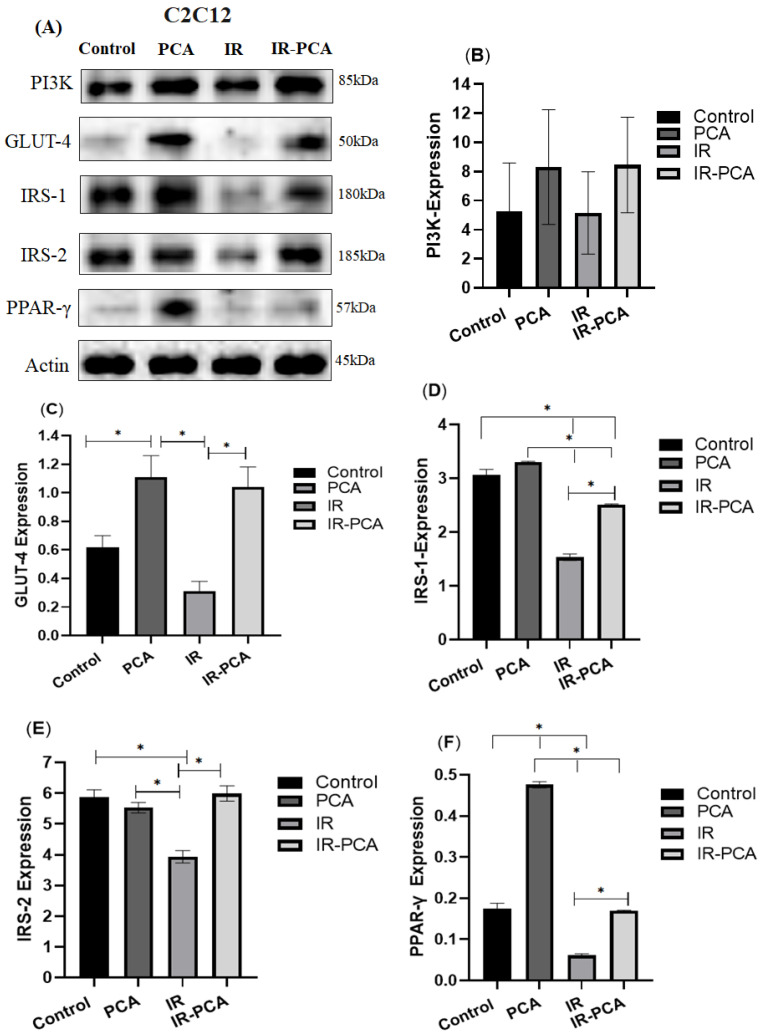
Effect of protocatechuic acid (PCA) on the protein expression of PI3K, GLUT4, IRS-1, IRS-2, and PPAR-γ proteins in C2C12 myotubes. (**A**) Immunoblot analysis, (**B**) level of PI3K expression, (**C**) level of GLUT-4 expression, (**D**) level of IRS-1 expression, (**E**) level of IRS-2 expression, (**F**) level of PPAR-γ expression. Data are shown as Mean ± S.D. of duplicate experiments. ANOVA with multiple comparisons was performed and statistical significance was set at *p* ≤ 0.05. * Statistically significant (*p* ≤ 0.05).

**Figure 8 ijms-24-09490-f008:**
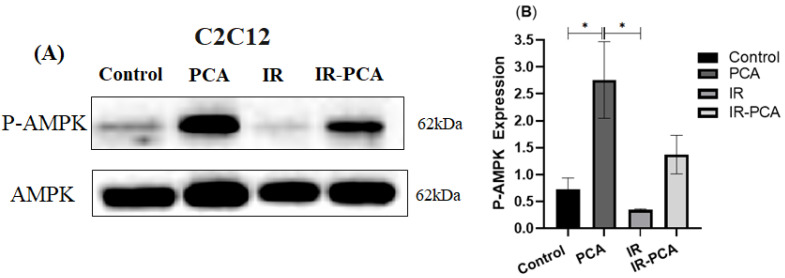
Effect of protocatechuic acid (PCA) on the protein expression of P-AMPK proteins in C2C12 myotubes. (**A**) immunoblot analysis, (**B**) level of P-AMPK expression. Data are shown as Mean ± S.D. of duplicate experiments. ANOVA with multiple comparisons was performed and statistical significance was set at *p* ≤ 0.05. * Statistically significant (*p* ≤ 0.05).

**Figure 9 ijms-24-09490-f009:**
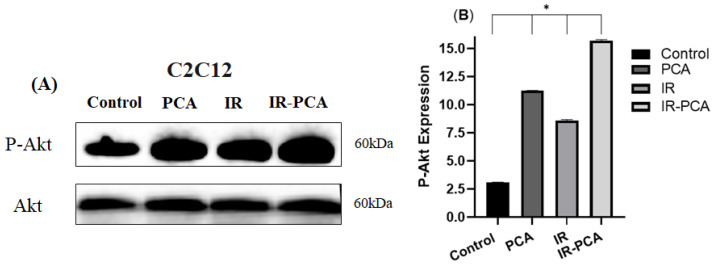
Effect of protocatechuic acid (PCA) on the level of expression of P-Akt proteins in C2C12 myotubes. (**A**) immunoblot analysis, (**B**) level of P-Akt expression. Data are shown as Mean ± S.D. of duplicate experiments. ANOVA with multiple comparisons was performed and statistical significance was set at *p* ≤ 0.05. * Statistically significant (*p* ≤ 0.05).

**Figure 10 ijms-24-09490-f010:**
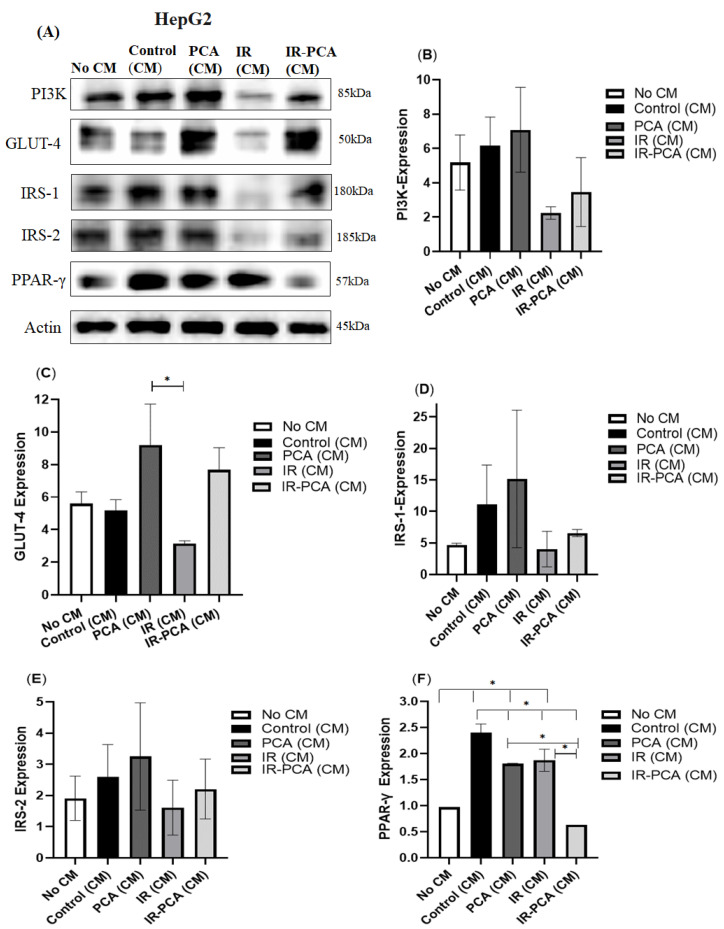
Effect of protocatechuic acid (PCA) on the protein expression of PI3K, GLUT4, IRS-1, IRS-2, and PPAR-γ proteins in HepG2. (**A**) Immunoblot analysis, (**B**) level of PI3K expression, (**C**) level of GLUT-4 expression, (**D**) level of IRS-1 expression, (**E**) level of IRS-2 expression, (**F**) level of PPAR-γ expression. Data are shown as Mean ± S.D. of duplicate experiments. ANOVA with multiple comparisons was performed and statistical significance was set at *p* ≤ 0.05. * Statistically significant (*p* ≤ 0.05). Conditioned Media (CM).

**Figure 11 ijms-24-09490-f011:**
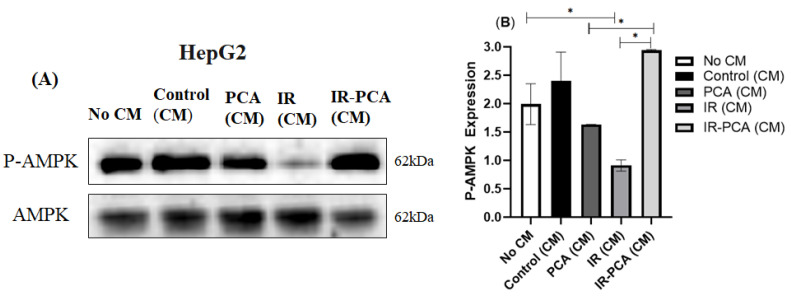
Effect of protocatechuic acid (PCA) on the protein expression of P-AMPK proteins in HepG2. (**A**) immunoblot analysis, (**B**) level of P-AMPK expression. Data are shown as Mean ± S.D. of duplicate experiments. ANOVA with multiple comparisons was performed and statistical significance was set at *p* ≤ 0.05. * Statistically significant (*p* ≤ 0.05). Conditioned Media (CM).

**Figure 12 ijms-24-09490-f012:**
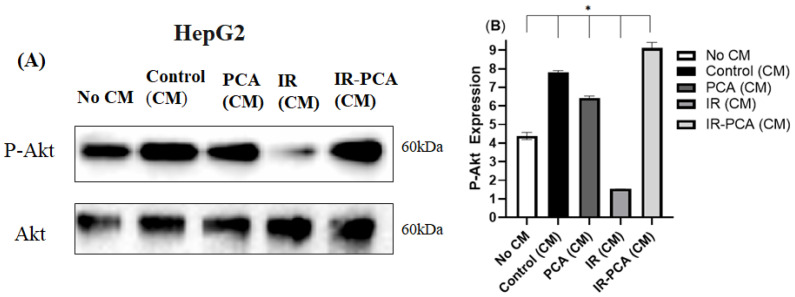
Effect of protocatechuic acid (PCA) on the protein expression of P-Akt proteins in HepG2. (**A**) immunoblot analysis, (**B**) level of P-Akt expression. Data are shown as Mean ± S.D. of duplicate experiments. ANOVA with multiple comparisons was performed and statistical significance was set at *p* ≤ 0.05. * Statistically significant (*p* ≤ 0.05). Conditioned Media (CM).

**Figure 13 ijms-24-09490-f013:**
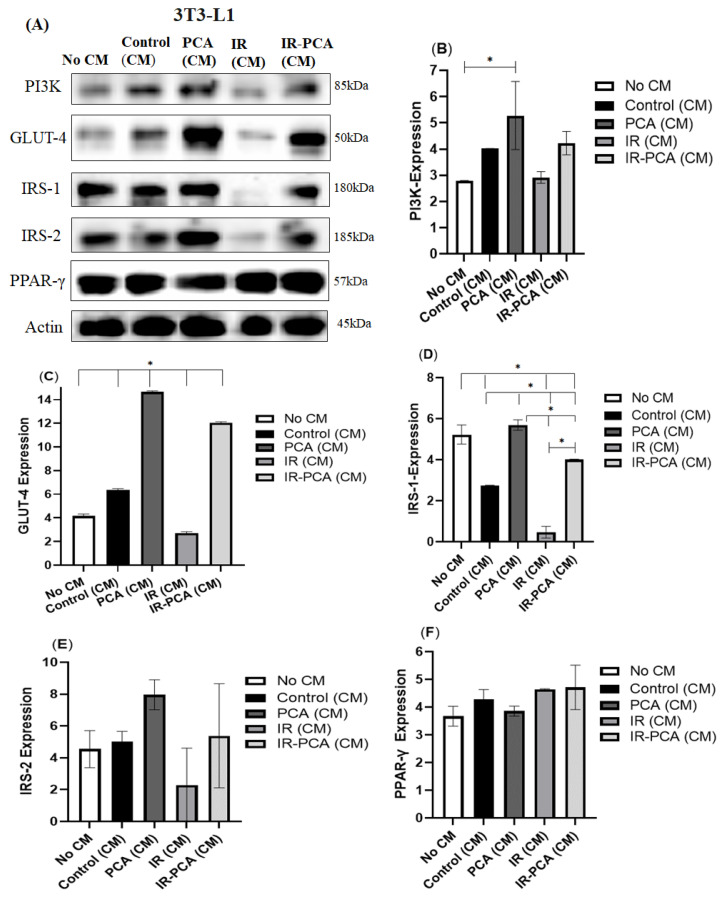
Effect of protocatechuic acid (PCA) on the protein expression of PI3K, GLUT4, IRS-1, IRS-2, and PPAR-γ proteins in 3T3-L1 adipocytes. (**A**) Immunoblot analysis, (**B**) level of PI3K expression, (**C**) level of GLUT-4 expression, (**D**) level of IRS-1 expression, (**E**) level of IRS-2 expression, (**F**) level of PPAR-γ expression. Data are shown as Mean ± S.D. of duplicate experiments. ANOVA with multiple comparisons was performed and statistical significance was set at *p* ≤ 0.05. * Statistically significant (*p* ≤ 0.05). Conditioned Media (CM).

**Figure 14 ijms-24-09490-f014:**
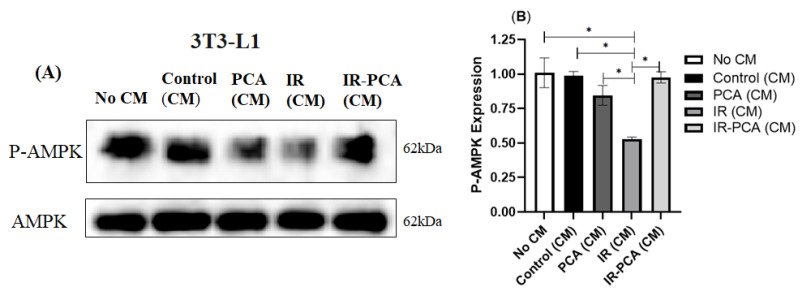
Effect of protocatechuic acid (PCA) on the protein expression of P-AMPK proteins in 3T3-L1 adipocytes. (**A**) immunoblot analysis, (**B**) level of P-AMPK expression. Data are shown as Mean ± S.D. of duplicate experiments. ANOVA with multiple comparisons was performed and statistical significance was set at *p* ≤ 0.05. * Statistically significant (*p* ≤ 0.05). Conditioned Media (CM).

**Figure 15 ijms-24-09490-f015:**
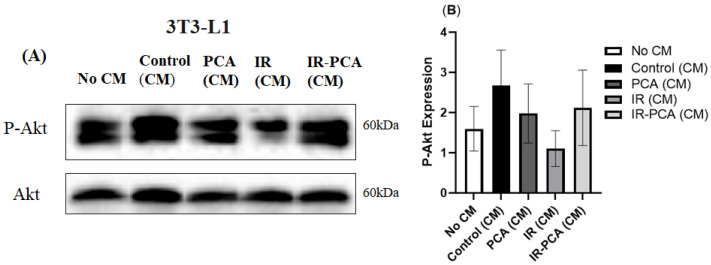
Effect of protocatechuic acid (PCA) on the protein expression of P-Akt proteins in 3T3-L1 adipocytes. (**A**) immunoblot analysis, (**B**) level of P-Akt expression. Data are shown as Mean ± S.D. of duplicate experiments. ANOVA with multiple comparisons was performed and statistical significance was set at *p* ≤ 0.05. Conditioned Media (CM).

**Figure 16 ijms-24-09490-f016:**
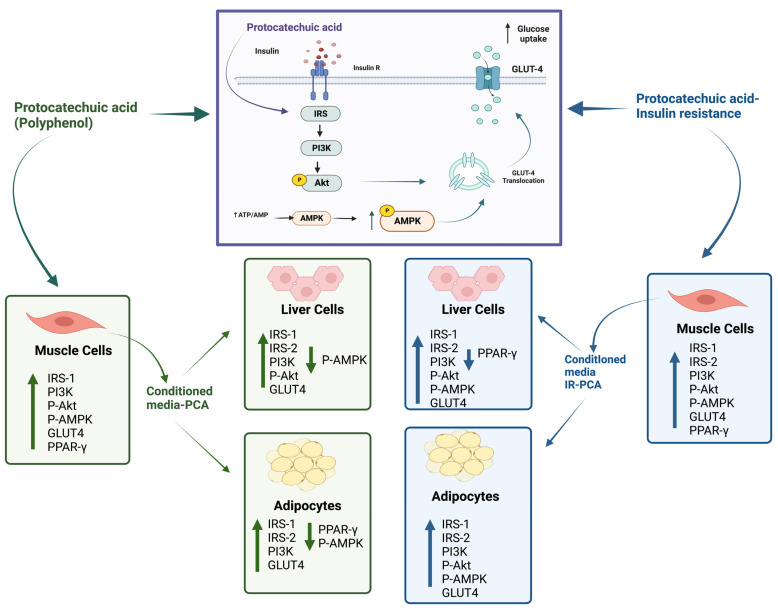
Summary of the research findings. Protocatechuic acid (PCA), Insulin resistance (IR).

## Data Availability

The datasets generated and analyzed during the current study are available from the corresponding authors upon reasonable request.

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
