# Peer review of "The Muscle-Conditioned Medium Containing Protocatechuic Acid Improves Insulin Resistance by Modulating Muscle Communication with Liver and Adipose Tissue"

_ijms, 2023, doi:10.3390/ijms24119490_

Round 1

Reviewer 1 Report

Dear Editor,

thanks so much for the opportunity to revise the work entitled "The muscle-conditioned medium containing Protocatechuic acid improves insulin resistance by modulating muscle communication with the liver and adipose tissue”.

The study shows that PCA strengthens insulin signaling by activating key proteins of that pathway and regulating glucose uptake. Further, conditioned media modulate crosstalk between muscle with liver and adipose tissue, thus regulating glucose metabolism.

The paper is well written, shows new insights in the problem of increasing number of people with diabetes mellitus. Diabetes Mellitus is a public health concern, affecting 10.5% of the population. All new ideas concerning methods to control impaired glucose metabolism is of great importance. This study investigated the role of PCA in improving insulin resistance and the crosstalk between muscle with liver and adipose tissue.

I have no specific remarks. Maybe the discussion might be more detailed, especially the part concerning AMPK.  

Thanks.

Author Response

Thank you for your valuable input

Reviewer 2 Report

Remarks on the paper submitted:

- Why is "Protocatechuic acid" written with capitals? It is not justified, please, correct. The same applies for the keywords.

- L60-61 Please, reformulate: " In the presence of insulin resistance, glucose uptake reduces by peripheral tissues [17]."

- L85-86: PCA is not an anthocyanin- please, correct.

- P5, Fig. 5: Why is glucose uptake higher in the presence of insulin in the IR group than in control group? The same question for these two groups without insulin (Figure 6). Are these differences significant?

- References should be double-checked (capital letters for journal names are missing).

Author Response

Thank you for your valuable input

Reviewer 3 Report

This study aimed to explore the effect of PCA exposure on myotubes (C2C12) and the effect of muscle secretome (conditioned medium) on adipose cells (3T3-L1) and hepatocytes (HepG2). Several complementary experiments have been performed to assess insulin resistance/glucose uptake in the three cell-lines.

The article is difficult to follow. The introduction is not logically structured. Moving figure 1 from Methods to Results and start with brief introduction of study design, and grouping the results per cell-line would improve readability. Conclusion should follow the discussion. A figure summarizing the main effects on skeletal muscle, liver and adipose tissue derived cells would be very helpful. The use of three levels of statistical significance also does not help clarity either. Moreover, many experiments state “duplicate experiment”, is no independent validation experiment performed?

The discussion contains a number of comparisons of individual results to other studies, but does not clearly addresses the research question regarding PCA effect on insulin resistance and communication between the “tissues”. 

Contradicting results regarding PPARG expression (#351) are observed and authors suggest that this might be caused by inhibition of adipocyte differentiation. This should be a control experiment, same for C2C12 myotube formation.

#134 and Figure 5 control conditioned medium increased glucose uptake in HepG2 by ~5 fold. In contrast, GLUT-4 (figure 10) is unchanged. This should be discussed.

The title is not substantiated by the conclusions of this study.

Specific remarks/suggestions

#9 specify T2DM

#51 skeleton muscle -> skeletal muscle

#60-61 rephrase sentence.

#69-71 rephrase sentence.

Grammar and word choice should be checked by a native English speaker.

Full gene name is not explained for all abbreviations at first use.

2.1. Add SD to percentage decrease, also in 2.6.

Figure 5. All comparisons to “No CM” with ** and ***, make it difficult to assess the significant changes between control, PCA and IR CM.

Figure 9b. Are PCA and IR not significantly changed compared to control?

Figure 13c, only No CM vs. IR-PCA significantly changed?

#333. “In C2C12 myotubes, IRS-1 expression increased by 7% in PCA vs. 332 Control, while in IR-PCA, the increase was significantly (p ≤ 0.001) greater vs. IR.” Add percentage increase for IR-PCA.

Methods: Indicate cell number correctly, e.g. (1 x 10E5 cell/well)

Clearly state how many replicates per experiment were analysed and how many independent experiments were performed.  Regarding cell viability, #444 “All experiments were performed in quadruplicate” and  #101 “Four independent experiment”, meaning total 16 samples per condition?

Author Response

Thank you for your valuable comments

Round 2

Reviewer 3 Report

Clarity and readability of the article has improved a lot. Discussion and conclusion are more to the point.

Few suggestions/remarks:

In study in 2.1 study design, include aim(s) of the study.

line 375 reference is missing for newly added data.

Check manuscript for consistence regarding reporting of SD. SD is not added everywhere and when reported (e.g. line 273, 302), check consistency regarding number of decimals of mean and sd, e.g. line 180: "15.583 ± 0.97 pmol without insulin"

Conclusion:

Sentence line 650-651 is unclear

Move sentence that starts in middle of line 655"(Incubating.... to no CM) before 652 "This study...

Author Response

Thank you for your valuable comments.
